# Personalized immunoglobulin aptamers for detection of multiple myeloma minimal residual disease in serum

Claudia Tapia-Alveal [1], Timothy R. Olsen[1] & Tilla S. Worgall [1✉]

Multiple myeloma (MM) is a neoplasm of plasma cells that secrete patient specific monoclonal immunoglobulins. A recognized problem in MM treatment is the early recognition of minimal residual disease (MRD), the major cause of relapse. Current MRD detection methods (multiparameter flow cytometry and next generation sequencing) are based on the analysis of bone marrow plasma cells. Both methods cannot detect extramedullary disease and are unsuitable for serial measurements. We describe the methodology to generate high affinity DNA aptamers that are specific to a patient's monoclonal Fab region. Such aptamers are 2000-fold more sensitive than immunofixation electrophoresis and enabled detection and quantification of MRD in serum when conventional MRD methods assessed complete remission. The aptamer isolation process that requires small volumes of serum is automatable, and Fab specific aptamers are adaptable to multiple diagnostic formats including point-of-care devices.

[1] Department of Pathology and Cell Biology, Columbia University Irving Medical Center, New York, NY 10032, USA. ✉email: tpw7@cumc.columbia.edu

Multiple myeloma (MM) is a plasma cell neoplasm that affects at least 130,000 patients worldwide[1,2]. MM disease status is monitored by the amount of monoclonal immunoglobulin (M-Ig) circulating in serum and the presence of malignant plasma cells in the bone marrow. The increased survival of patients with MM has directed the focus on early detection of minimal residual disease (MRD). Several studies showed that patients with negative MRD status attain superior clinical outcomes[3–7]. These findings prompted the international myeloma working group (IMWG) to include MRD detection as a measure of response[8] and regulatory agencies, Federal Drug Administration (FDA) and the European Medicines Agency (EMA), to issue recommendations regarding the use of MRD for clinical endpoints[9,10]. The current state-of-the-art methods to

**Fig. 1 Isolation and validation of daratumumab aptamer aptD. a** Left: Target and counter-targets antibodies were immobilized on protein G columns to optimize exposure of Fab to ssDNA. Right: SELEX approach. During selection (1) immobilized target (e.g., daratumumab) was incubated with a randomized 40-*mer* ssDNA oligonucleotide SELEX library. Non-binding ssDNA were washed off while binding sequences were retained and eluted for incubation with counter-targets (e.g., polyclonal antibodies, beads) (2, 'Counter selection') Flow-through was collected, PCR amplified (3), strand-separated (4, 'ssDNA recovery') and incubated with target in the next SELEX round. **b** Left: Nucleotide sequence and 2D structure prediction for aptD at 25 °C, PBS, 2 mM MgCl₂: $\Delta G = -7.62$ kcal mol$^{-1}$. Right: Specificity. Daratumumab and control IgGs rituximab, M-Ig from three unrelated MM patients and polyclonal IgG were immobilized on protein G coated plates and incubated for 30 min with biotin-aptD or controls (biotin-scrambled aptamer (scr), no aptamer (no apt)). Binding was detected with streptavidin-HRP and visualized with TMB chromogenic substrate (370 nm). **c** Limit of detection. Biotin-aptD, immobilized on streptavidin coated plates, was incubated with serum containing decreasing concentrations of daratumumab. Six independent measurements performed in triplicates determined a limit of detection of $4.5 \times 10^{-6}$ g dl$^{-1}$. **d** Binding affinity ($K_D$) by surface plasmon resonance (SPR). Biotin-aptD was immobilized on SwitchAvidin sensors, exposed to increasing concentrations (6.25–200 nM) of daratumumab or control polyclonal human IgG (dotted line, representative 100 nM) and graphed after subtracting results obtained with control (biotin-scrambled aptD) analyzed in a parallel channel. Sensograms show aptD specific binding that determined a dissociation constant ($K_D$) of 2.67 nM. **e** Specificity and sensitivity for daratumumab in serum by ELONA. Biotin-aptD, immobilized on streptavidin coated plates was incubated with serum from three patients (D1-D3) obtained before (pre-treatment) and after daratumumab treatment (treatment) at different dilutions (dil x1000). Biotin-scrambled aptD (scr) was used as a control with pre-treatment and treatment samples. No aptamer (no apt) control was included for treatment samples only. Sera were diluted up to 1:40,000 with buffer. No significant signal was detected in pre-treatment samples or with scrambled control (scr) or no aptamer (no apt). AptD detected daratumumab in all sera obtained after treatment at all dilutions. **f** Specificity and sensitivity in serum by homogenous time-resolved fluorescence (HTRF). Insert: HTRF measures fluorescence resonance energy transfer (FRET) that occurs when aptD labeled with terbium (streptavidin-terbium (SA-Tb), ex. 340/em. 620 nm) is in proximity of acceptor fluorophore d2 labeled anti-human IgG (anti IgG-d2; ex. 620/em. 665 nm). Graph: FRET occurs when aptD [biotin-aptD-SA-Tb] was incubated with treatment serum (patient D1), but not when incubated with pre-treatment serum (Pre-Tx) or control sera (MM, no-MM), nor when controls (scrambled aptD, no apt) were incubated with treatment serum. All data represent means of at least $n = 3$ biologically independent samples ± SD, two-tailed Student *t*- test. **p < 0.01; ***p < 0.001; ****p < 0.0001, n.s. not significant.

detect MRD, multiparametric flow cytometry (MPFC) and next-generation sequencing (NGS), analyze bone marrow-derived plasma cells. Both methods are very sensitive and can detect one abnormal plasma cell in $10^5$ (MPFC) or $10^6$ (NGS) background cells. Recognized shortcomings are the inability to detect extramedullary disease and the unsuitability for serial measurements[6,7,11–13]. Additional pre-analytical challenges include poor survival of aspirated plasma cells and non-representativeness of samples secondary to non-uniform marrow infiltration or a suboptimal aspiration procedure[14–17].

MRD detection in serum would facilitate monitoring of relapse, help predict outcomes, and could provide decentralized care if an immunocompromised patient population is advised not to visit hospitals for routine check-ups[18]. Current methods to measure M-Ig in serum are not sensitive enough to detect MRD. The limit of detection (LoD) for M-Ig by serum protein electrophoresis (SPEP), the standard method for initial detection and follow-up of monoclonal gammopathies is 0.1 g dl$^{-1}$. Immunofixation electrophoresis (IFE) that is 10-fold more sensitive than SPEP (LoD, 0.01 g dl$^{-1}$) is used to distinguish a very good response (VGPR), where M-Ig is visible, from complete remission (CR) where M-Ig is not detected[2]. Mass spectrometry is 100-fold more sensitive than IFE (LoD 0.0001 g dl$^{-1}$) but is not suitable for MRD detection because of its inaccuracy in the presence of endogenous polyclonal immunoglobulin levels over 0.8 g dl$^{-1}$ (i.e., within normal range)[19–22].

The MM M-Ig has two domains[23]: (1) a crystallizable fragment (Fc) that defines the antibody isotype, and (2) an antigen binding fragment (Fab). The Fab region is unique to the patient's M-Ig and can therefore be used as a personal biomarker to detect the presence of disease and to measure a patient's tumor burden.

Aptamers are in-vitro selected, single-stranded oligonucleotides with unique three-dimensional structures that confer high affinity and specificity towards their target. They can be isolated by systematic evolution of ligands by exponential enrichment (SELEX) from single-stranded oligonucleotide libraries that contain a large number ($10^{14}$) of unique sequences[24–27]. Once identified, aptamers can be easily and reliably reproduced using established DNA synthesis techniques that are, in contrast to monoclonal antibodies, unaffected by batch-to-batch

variability[28,29]. Validated aptamers can be used for molecular recognition of biomarkers and can further be adapted to standard clinical laboratory platforms (i.e., immunoanalyzers) and point-of-care devices.

Herein we describe the isolation of M-Ig-specific and high-affinity ($K_D < 10$ nM) DNA aptamers that were 2000-fold (LoD $4.5 \times 10^{-6}$ g dl$^{-1}$) more sensitive than serum protein IFE (LoD $1 \times 10^{-2}$ g dl$^{-1}$). The M-Ig aptamers detected residual disease in serum at a time when complete remission was diagnosed by conventional methods. Personalized high-affinity aptamers provide the sensitivity to monitor medullary and extramedullary MRD in serum and are suitable for frequent, long-term monitoring.

## Results

**Isolation and characterization of a daratumumab specific aptamer.** The overall strategy was to first isolate target-specific aptamers by eliminating binders to common IgG features or beads with two counter-selection steps per SELEX round (Fig. 1a left), and then to select aptamers that were most sensitive in serum. The anti-CD38 therapeutic monoclonal antibody (mAb) daratumumab that is approved for MM therapy[30–32] was the first target and was used to establish the protocol. We chose this IgG Kappa because IgG Kappa represents the most prevalent M-Ig isotype in MM and serum samples from MM patients obtained pre- and post daratumumab treatment provided the relevant validation matrix. Advantageous for method development was furthermore the commercial availability of the pure form.

For SELEX, daratumumab was immobilized on protein G beads to expose the Fab to the randomized oligonucleotide SELEX library. Selection rounds included counter-selections with polyclonal IgG and empty beads (Fig. 1a, Supplementary Fig. 1a). SELEX stages (e.g., washes and elution) were monitored by PCR (Supplementary Fig. 1c). At the end of a SELEX round the recovered sequences were pooled and PCR amplified (Supplementary Fig. 1c, d). The decrease of PCR cycles required to generate sufficient DNA for the next SELEX round was used to evaluate enrichment[33]. After six SELEX rounds binding sequences (Supplementary Fig. 1d) were PCR cloned using the

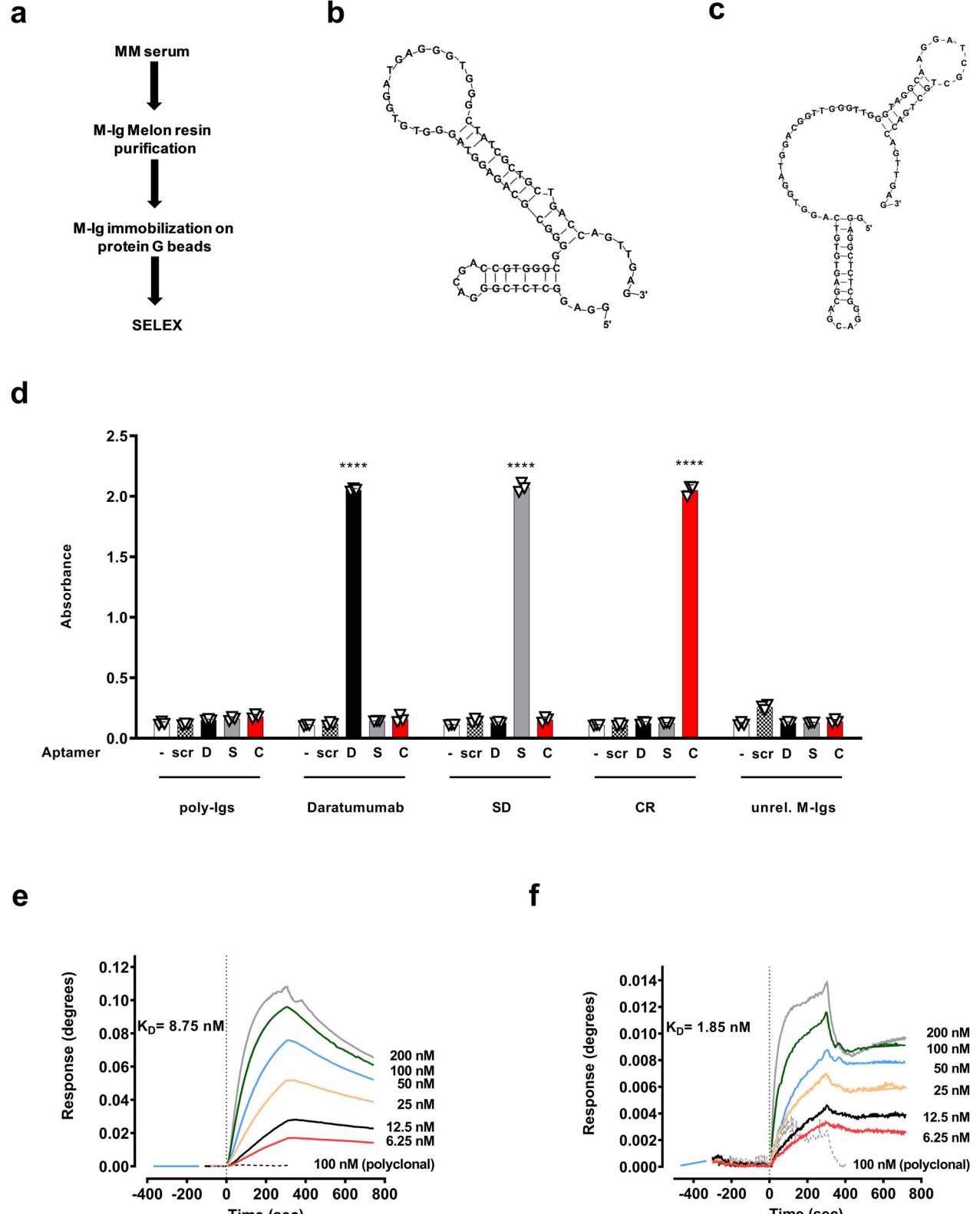

TOPO TA cloning kit and 20 clones were sequenced. Sequences were tested by ELONA and, based on results obtained for binding to daratumumab in buffer and serum, distributed into two families (Supplementary Fig. 1e). Here, we focus on one of them, aptD, that bound to daratumumab in serum (Fig. 1b, left, nucleotide sequence and 2D structure).

We assessed aptD specificity by testing cross-reactivity with various IgGs on ELONA. AptD showed linear binding to daratumumab from 5 to 15 min after addition of TMB substrate and no significant binding to therapeutic mAb rituximab, polyclonal IgG, or three different IgG Kappa M-Igs purified from MM patient sera (Fig. 1b, right). No binding was detected

**Fig. 2 Isolation and validation of patient M-Ig aptamers. a** Patient M-IgG, one from a patient with stable disease (SD) and one from a patient who had achieved complete remission (CR), were purified from serum by Melon-resin columns and then immobilized on protein G columns for SELEX. **b–c** Nucleotide sequence and 2D structure prediction at 25 °C on PBS, 2 mM MgCl$_2$ for aptS that was isolated after 7 SELEX rounds for the M-Ig of patient SD ($\Delta G = -7.21$ kcal mol$^{-1}$) (**b**), and for aptC that was isolated after 6 SELEX rounds for the M-Ig of patient CR ($\Delta G = -8.94$ kcal mol$^{-1}$) (**c**). **d** Specificity of aptS and aptC. Polyclonal immunoglobulins (poly-Igs), daratumumab, M-Ig from patient SD (SD), M-Ig from patient CR (CR), and M-Igs from unrelated patients (unrel. M-Igs) were immobilized on protein G-coated plates and incubated for 30 min with no aptamer (-), biotin-labeled scrambled aptamers (scr), biotin-labeled aptD (D), biotin-labeled aptS (S), or biotin-labeled aptC (C). Binding was detected with streptavidin-HRP and visualized with TMB (370 nm). AptS, aptC, and aptD were specific and did not cross-react. **e–f** Binding affinity by surface plasmon resonance (SPR). Biotin-labeled aptS or aptC were immobilized on SwitchAvidin sensors, exposed to increasing concentrations (6.25–200 nM) of target M-Ig, or control polyclonal human IgG (dotted line, representative 100 nM). Results show specific binding that was obtained after subtracting control (biotin-scrambled aptamer) that was analyzed in parallel. Sensograms show dissociation constant ($K_D$) of 8.75 nM for aptS (**e**) and 1.85 nM for aptC (**f**). No significant signal was observed for polyclonal IgG control (dotted line, 100 nM). All data represent means of at least $n = 3$ biologically independent samples ± SD, two-tailed Student $t$- test. **$p < 0.01$; ***$p < 0.001$; ****$p < 0.0001$.

with a scrambled aptamer (scr) or no aptamer control (no apt). The LoD for aptD, determined by measuring serum containing decreasing concentrations of daratumumab was $4.5 \times 10^{-6}$ g dl$^{-1}$ (Fig. 1c) which was 2000-fold lower than the LoD for immunofixation $1 \times 10^{-2}$ g dl$^{-1}$. AptD did not bind to daratumumab Fc fragments (Supplementary Fig. 2a) but bound to (Fab')$_2$ (Supplementary Fig. 2b). These results indicate that Fc binding sequences were successfully eliminated in our method. SPR is the state-of-the-art method to evaluate real-time interactions between aptamers and targets in a sensitive and label independent manner. Binding affinity ($K_D$) was assessed between biotin-aptD immobilized on a SwitchAvidin biosensor chip and daratumumab in solution. A biotin-scrambled aptamer in which the sequence between primers is randomized was used in parallel to control for non-specific interaction. This signal was subtracted from the aptD signal in the sensograms. Kinetic analysis of aptD binding to daratumumab determined a $K_D$ of 2.67 nM (Fig. 1d). There was no significant binding of aptD to polyclonal Igs. The results demonstrate the feasibility of generating an aptamer with a high specificity and affinity to a selected target IgG mAb.

**Detection of daratumumab in serum with aptD.** The detection of molecules in serum can be affected by the presence of matrix inherent interferences and modifications that the target acquired during circulation in the bloodstream. We tested the ability of aptD to detect daratumumab in sera obtained from three patients before and after daratumumab treatment for at least 6 months (D1–D3). Of note is that the patients are being treated for MM and that their M-Igs are present and detectable in the samples. Immobilized aptD detected daratumumab in the ELONA pull-down assay in all post-treatment sera and in all tested dilutions (up to 1:40,000) but not in pre-treatment sera (Fig. 1e). No appreciable response was detected with controls (biotin-aptD scrambled (scr) and no aptamer (no apt)). Thus, aptD is specific for daratumumab and sensitive over a wide concentration range in serum. Because ELONA is a heterogeneous assay in which repeated washes remove potential serum interferences, we tested the ability of the aptamer to detect daratumumab in a homogeneous setting. The HTRF format combines fluorescence resonance energy transfer (FRET) with time-resolved measurements that eliminates short-lived background fluorescence[34–37]. The HTRF ratio (665/620) measures FRET that occurs when two fluorophores (i.e., donor terbium-cryptate (Tb), and acceptor allophycocyanin (d2)) are in proximity. Serum obtained before and after daratumumab treatment was incubated with a terbium complexed-aptD [biotin-aptD + streptavidin-Tb (SA-Tb)] and anti-IgG-Fc labeled with d2 (α IgG-d2) (Fig. 1f, insert). FRET was observed when Tb-labeled aptD was incubated with serum obtained after daratumumab treatment (treatment) but not when

this serum was incubated with controls (biotin-aptD scrambled (scr), no apt). (Fig. 1f). FRET was not observed when terbium-labeled aptD was incubated with serum obtained before treatment (pre-tx) or control sera (MM, no MM). Results support that aptD does not bind to a serum molecule other than daratumumab.

**Identification of personalized aptamers for MM patients.** Next, we applied the protocol to isolate M-Ig specific aptamers for one patient with stable disease (SD) and one patient who had achieved complete response (CR) after ASCT, and relapse 25 months later. M-Igs were purified from serum by using Melon gel IgG purification resins (Supplementary Fig. 3a) followed by immobilization on protein G beads for SELEX (Fig. 2a). Recovered sequences were PCR amplified and the number of PCR cycles required to amplify at least 80 pmol for the next SELEX round were recorded to assess progress (Supplementary Fig. 3b, c). Seven rounds were done for SD M-Ig SELEX (Supplementary Fig. 3b). From the identified sequences ($n = 19$) 37% were the same aptamer and the only one that bound M-Ig that had been spiked into serum on ELONA. It was ultimately named aptS (Fig. 2b, Supplementary Fig. 3d). CR M-Ig SELEX required six rounds and identified aptC (Fig. 2c) in 10% of the selected sequences ($n = 19$) and was the only one that bound the target in serum (Supplementary Fig. 3c, e). Specificity testing by ELONA showed that aptS and aptC did not cross-react with polyclonal immunoglobulins, daratumumab, or unrelated mAbs immobilized on protein G-coated plates (Fig. 2d). Absorbance with scrambled aptamers was not significantly higher than no apt controls, indicating that the primer regions were not directly involved in binding. SPR analyses including scrambled controls revealed a $K_D$ of 8.75 nM for aptS (Fig. 2e) and a $K_D$ of 1.85 nM for aptC (Fig. 2f). Scrambled aptamers did not bind to M-Ig from either patient, and neither aptS nor aptC bound significantly to polyclonal IgG. These findings are consistent with the high specificity and affinity of aptC and aptS.

**Sensitivity of personalized M-Ig aptamer aptS in serum.** First, the available sequential SD serum samples obtained at diagnosis (0 months), at 4, and at 18 months were analyzed by SPEP to determine M-Ig concentrations (Fig. 3a, top) and by IFE to confirm the IgG isotype (Fig. 3a, bottom). The same sera were then analyzed with aptS in the ELONA pull-down format up to a dilution of 1:40,000. AptS detected M-Ig in all dilutions. No significant signal was observed with controls (aptS-scrambled, no apt), or when aptS was incubated with serum controls (MM, no MM) (Fig. 3b). We next assessed binding in the HTRF assay to rule out that the washing steps in ELONA would have eliminated interferences. SD sera containing SA-Tb and α IgG-d2 were incubated with biotin-aptS, or controls (aptS-scrambled (scr), no

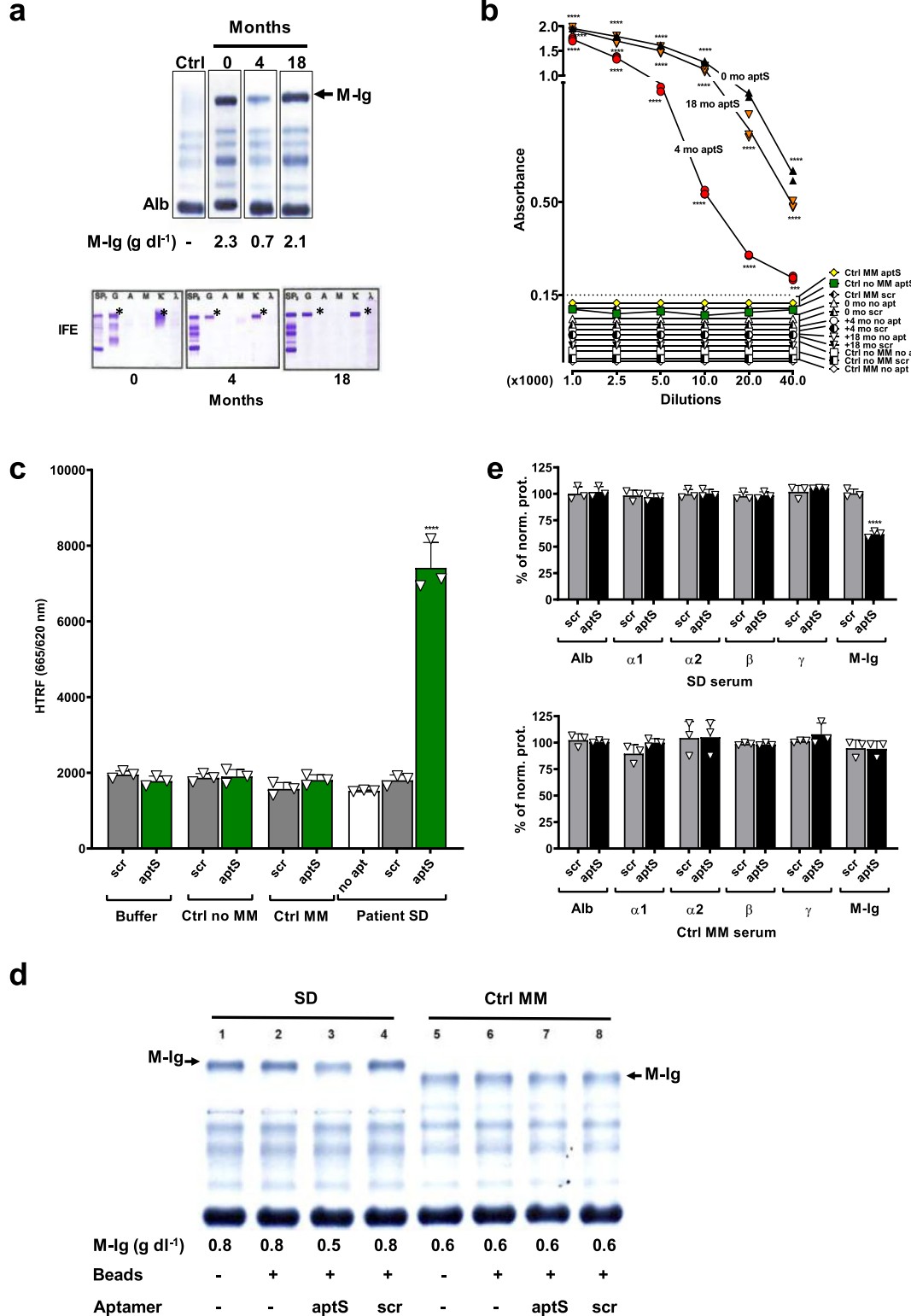

apt) (Fig. 3c). FRET was only observed for the condition where aptS was incubated with patient SD serum. Control conditions that tested scrambled aptamer, no apt, or control sera (no MM, MM) did not generate FRET (Fig. 3c). Next, to rule out that aptS binds to a protein in patient serum that is not the target M-Ig we performed pull-down assays in which the serum sample was incubated with biotin-aptS or controls (biotin-aptS scrambled (scr), no apt) before binding to streptavidin agarose beads.

Collected flow-through was electrophoresed by SPEP and proteins quantified by densitometry. Results show that patient M-Ig decreased from $0.8 \, \text{g dl}^{-1}$ to $0.5 \, \text{g dl}^{-1}$ in the sample with biotin-aptS (Fig. 3d, lane 3) but not in the samples incubated with controls (beads only, aptS-scrambled (scr); Fig. 3d, lanes 2 and 4). No pull-down was observed when aptS and controls were incubated with a MM control serum (Fig. 3d, lanes 5–8). Upon normalization, we found a 37.5 % decrease of SD M-Ig and no

**Fig. 3 Specificity and sensitivity of aptS in patient serum. a** Serum protein electrophoresis (SPEP, top) and immunofixation (IFE, bottom), composite of chronological data. Commercial serum assay control (Ctrl) and sequential serum samples (0, 4, 18 months) obtained from patient SD were analyzed by SPEP. The M-Ig migrates in the cathodal gamma fraction (arrow). Concentrations were 2.3 g dl$^{-1}$ (0 months), 0.7 g dl$^{-1}$ (4 months), and 2.1 g dl$^{-1}$ (18 months). IFE identified the M-Ig as an IgG Kappa in each sample (asterisks). SP = serum protein, no immunofixation, G = SP fixed with anti-IgG antibody (Ab), A = SP fixed with anti-IgA Ab, M = SP fixed with anti-IgM Ab, K = SP fixed with anti-Kappa light chain Ab, λ = SP fixed with anti-Lambda light chain Ab. **b** Specificity and sensitivity in serum by ELONA. Biotin-aptS and controls (biotin-scrambled (scr), no aptamer (no apt)) were immobilized on streptavidin-coated plates and incubated for 30 min with increasing dilutions (1:1000–1:40,000) of patient SD sera that had been analyzed by SPEP. Serum from 0, 4, and 18 months or controls from 10 patients with or without MM (Ctrl MM, Ctrl no MM) were incubated with biotin aptS. Biotinylated scrambled aptamer (scr) and no aptamer (no apt) were included (0; 4; 18 months; control no MM and control MM). Binding was detected with goat anti-human IgG-HRP and visualized with TMB (370 nm). AptS was specific for SD serum and sensitive in all dilutions (tested up to 1:40,000). **c** Specificity and sensitivity in serum by HTRF. FRET occurs when aptS [biotin-aptS-SA-Tb] was incubated with SD serum, but not with control (no MM, MM, buffer), nor with no apt or scrambled controls (biotin-scr-SA-Tb) in any condition. **d** Pull-down of M-Ig from SD serum. Serum from SD (lanes 1–4) and MM control (lanes 5–8) were incubated with biotin-aptS (aptS) or biotin-aptS scrambled (scr) on streptavidin (SA) beads. Flow-through was analyzed by SPEP and protein fractions were quantified by densitometry. SD and Ctrl MM sera were incubated with buffer (1 and 5), SA-beads (2 and 6), aptS + SA beads (3 and 7) and scr + SA beads (4 and 8). Incubation with aptS reduced M-Ig from 0.8 g dl$^{-1}$ (lane 2) to 0.5 g dl$^{-1}$ (lane 3). No effect was observed with scr control (lane 4). AptS had no effect on M-Ig in control serum (lanes 5-8). **e** Effect of aptS on serum protein fractions (based on **d**). Serum protein fractions albumin (Alb), alpha 1 (α1), alpha 2 (α2), beta 1 and beta 2 (β), and polyclonal gammaglobulins (γ) were quantified after pull down and graphed as percentage of control (lanes 2 and 6 on **d**). SD serum (top): AptS 'pulled out' SD M-Ig but did not affect other SD serum protein fractions. Scrambled aptS (scr) had no effect. Control MM serum (bottom): AptS and scr control did not affect the concentrations of an unrelated M-Ig or serum protein fractions in control serum (Ctrl MM). All data represent means of at least $n = 3$ biologically independent samples ± SD, two-tailed Student $t$- test. $**p < 0.01$; $***p < 0.001$; $****p < 0.0001$.

evidence for a reduction of other serum proteins including the polyclonal gammaglobulin fraction (Fig. 3e, SD serum). Scrambled aptamer did not pull-down or affect serum protein fractions or M-Ig of patient SD (Fig. 3e, SD serum, scr) and aptS did not pull-down or affect serum protein fractions or M-Ig of a control MM patient (Fig. 3e, Ctrl MM serum). The results confirm specificity of aptS for the target M-Ig and illustrate the high-affinity binding to M-Ig directly in serum.

**Sensitivity of personalized M-Ig aptamer aptC in serum.** Next, we analyzed sequential serum samples from the CR patient obtained pre-ASCT (0 months), throughout CR (6–22 months) and at relapse (25 months) by SPEP (Fig. 4a, top) and IFE (Fig. 4a, bottom). The M-Ig that migrates in the gamma region was quantifiable pre-ASCT (2.9 g dl$^{-1}$, 0 months) and at relapse (0.5 g dl$^{-1}$, 25 months). At 25 months the monoclonal band is barely visible by SPEP but clearly distinguishable by IFE. At 6–22 months when the patient was diagnosed to be in CR, SPEP and IFE were unremarkable and free Kappa light chains, the Kappa/Lambda ratio, albumin, beta2 microglobulin, LDH, quantitative IgGs, and polyclonal immunoglobulins were all within normal limits. MPFC was negative at 6 months (Table 1).

Analysis of the same sera with aptC by ELONA 'pull-down' detected M-Ig in all samples. In the pre-ASCT sample aptC detected M-Ig in all dilutions (tested up to 1:40,000) (Fig. 4b). No significant signal was obtained with controls (aptC scrambled (scr), no apt) in CR serum, or when aptC was incubated with control sera (no MM, MM). At 6–25 months M-Ig was detected by aptC in all samples up to a 1:16 dilution, and up to a 1:500 dilution in samples obtained at 22–25 months. No significant background signal was obtained with controls (aptC scrambled (scr), no apt) or control sera (no MM, MM) (Fig. 4c).

M-Ig detected during CR were quantified using samples with known concentrations as standards. Results obtained 6 months after ASCT detected M-Ig ($0.37 \times 10^{-3}$ g dl$^{-1}$) and thus residual disease. M-Ig concentrations increased exponentially after 12 months ($0.45 \times 10^{-3}$ g dl$^{-1}$ (12 months), $0.48 \times 10^{-2}$ g dl$^{-1}$ (14 months), $0.13 \times 10^{-1}$ g dl$^{-1}$ (17 months), $0.79 \times 10^{-1}$ g dl$^{-1}$ (22 months), and $4.8 \times 10^{-1}$ g dl$^{-1}$ (25 months) (Fig. 4d). Next, aptC was tested in the HTRF assay. FRET was only observed when aptC was incubated with patient CR serum. No significant FRET signal was obtained when aptC was incubated with control sera (no MM, MM), or when patient CR serum was incubated

with controls (aptC-scrambled (scr), or no apt) (Fig. 4e). To further confirm target specificity we performed pull-down assays where biotin-aptC and controls (biotin-aptC scrambled (scr), no apt) were incubated with patient or control sera prior to binding to streptavidin columns. The flow-through was electrophoresed and quantified by densitometry. AptC decreased CR M-Ig from 0.6 g dl$^{-1}$ to 0.3 g dl$^{-1}$ (Fig. 4f, lane 3) but had no effect on other serum protein fractions. No pull-down was observed with the scrambled or the no apt controls (Fig. 4f, lane 2 and 4). AptC had no effect on the serum fractions or M-Ig in control serum (MM) (Fig. 4f lane 7 and Fig. 4g Ctrl MM serum). Normalized data showed a 50% reduction CR M-Ig but no effect on other serum proteins including the polyclonal gammaglobulin fraction (Fig. 4g CR serum) confirming aptC specificity, high affinity and binding to CR M-Ig in serum. The results obtained for both patients demonstrate that personalized M-Ig aptamers are specific and suitable for sensitive and long-term follow up of serum M-Ig levels.

## Discussion

The goal of this study was to evaluate M-Ig specific aptamers for detection of MM MRD in patient serum. MRD is the major cause for MM relapse and early recognition affects treatment decisions and life expectancy[3,8,11]. State-of-the-art MRD detection methods analyze bone marrow plasma cells by MPFC and NGS. With input of the appropriate number of cells these methods achieve very high sensitivities. Recognized shortcomings however are the inability to assess extramedullary disease and, because of the invasiveness of bone marrow aspiration, their unsuitability for serial measurements.

We developed a SELEX method to isolate high-affinity aptamers that are specific to the Fab region of a patient's M-Ig. In our approach the M-Ig Fab region is a personal biomarker and an aptamer is generated for each patient. We chose the therapeutic monoclonal daratumumab (anti-CD38 IgG Kappa) as the initial target because IgG Kappa represents the most prevalent M-Ig isotype in MM, and because serum samples from MM patients obtained pre-and post daratumumab treatment provided the relevant validation matrix. The samples contain disease-specific or therapy-related molecules that could interfere with our assay, as well as daratumumab that had circulated for extended time during which it could have acquired modifications that could affect binding. The optimized method generated a highly sensitive

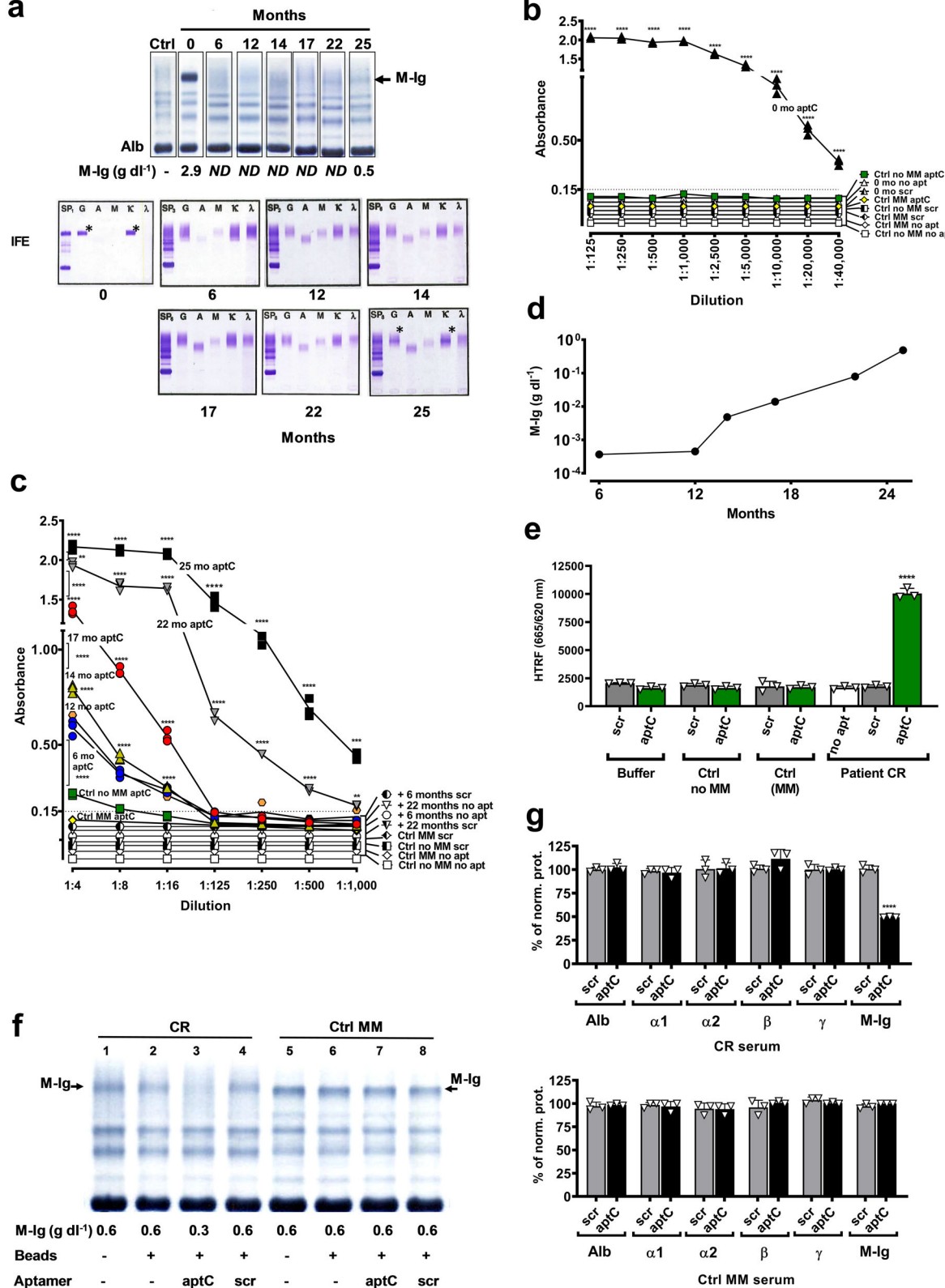

Fab-specific aptamer that detected daratumumab with a 2000-fold higher sensitivity than immunofixation and 20-fold higher sensitivity than mass spectrometry (Fig. 5).

Results obtained with aptS and aptC provide a proof-of-concept. The target specificity and high affinity ($K_D$ < 10 nM) supported the robustness of the protocol. The consistent

sensitivity in serum, including in the presence of high concentrations of polyclonal immunoglobulins that had affected use of mass spectrometry for detection of MRD, supports its clinical utility for M-Ig detection. AptS, generated for a patient with stable disease detected M-Ig in consecutive samples equivalent to SPEP and IFE. AptC, generated for a patient who had achieved

**Fig. 4 Specificity and sensitivity of aptC in patient serum. a** Serum protein electrophoresis (SPEP, top) and immunofixation (IFE, bottom), composite of chronological data. Control (Ctrl) and sequential M-Ig serum samples (0, 6, 12, 14, 17, 22, and 25 months) analyzed by SPEP show M-Ig in the cathodal gamma fraction (arrow) at 0 and 25 months at concentrations of 2.9 g dl$^{-1}$ (0 months, Pre-ASCT) and 0.5 g dl$^{-1}$ (25 months, relapse). M-Ig was not detected (ND) in months 6–22. IFE identified the M-Ig as an IgG Kappa (asterisks) at 0 months (pre-ASCT) and at 25 months. M-Ig was not detected by IFE in months 6–22. Alb (albumin), SP = serum protein (no immunofixation), G = SP fixed with anti-IgG antibody (Ab), A = SP fixed with anti-IgA Ab, M = SP fixed with anti-IgM Ab, K = SP fixed with anti-Kappa light chain Ab, λ = SP fixed with anti-Lambda light chain Ab. **b–c** Specificity and sensitivity in serum by ELONA. Patient CR serum obtained at 0 months and controls (MM, no MM) were diluted 1:125–1:40,000 (**b**). Patient CR sera obtained at months 6–25 and controls were diluted 1:4–1:1,000 (**c**). **b–c** 0 months, 6 months, 12 months, 14 months, 17 months, 22 months, 25 months and control sera (Ctrl MM, Ctrl no MM) were incubated for 30 min with biotin-labeled aptC immobilized on streptavidin-coated plates. Biotin-scrambled aptamer (scr) and no aptamer (no apt) controls were also included for: 0 months, control no MM, control MM, 6 months, and 22 months. Binding was detected with goat anti-human IgG-HRP and visualized with TMB (370 nm). Binding was observed in all conditions where aptC was incubated with patient CR serum. No signal was obtained when aptC was incubated with control sera, or when controls (aptC scrambled (scr), no apt) were incubated with patient or control sera. **d** Quantitation of residual disease in serum obtained during remission. M-Ig in remission sera **c** were quantified using samples with known concentrations (0 months, 25 months) as standards. Concentrations were $0.37 \times 10^{-3}$ g dl$^{-1}$ (6 months), $0.45 \times 10^{-3}$ g dl$^{-1}$ (12 months), $0.48 \times 10^{-2}$ g dl$^{-1}$ (14 months), $0.13 \times 10^{-1}$ g dl$^{-1}$ (17 months), $0.79 \times 10^{-1}$ g dl$^{-1}$ (22 months) to $4.8 \times 10^{-1}$ g dl$^{-1}$ at 25 months demonstrating exponential increase after 12 months. **e** Specificity and sensitivity in serum by HTRF. FRET occurs when aptC [biotin-aptC-SA-Tb] was incubated with CR serum, but not with controls (no MM, MM, buffer), nor with scrambled aptamer (scr) or no aptamer (no apt) in any condition. **f** Pull-down of M-Ig from CR serum. Serum from CR (lanes 1–4) and MM control (Ctrl MM: lanes 5–8) were incubated with biotin-aptC (aptC) or biotin-aptC scrambled (scr) on streptavidin (SA) beads. Flow-through was analyzed by SPEP and protein fractions were quantified by densitometry. CR and Ctrl MM sera were incubated with buffer (1 and 5), SA-beads (2 and 6), aptC + SA beads (3 and 7) or scr + SA beads (4 and 8). AptC reduced M-Ig in CR serum from 0.6 g dl$^{-1}$ (lane 1) to 0.3 g dl$^{-1}$ (lane 3). No effect was observed with scr control (lane 4). AptC had no effect on M-Ig in control serum (lanes 5–8). **g** Effect of aptC on serum protein fractions (based on **f**). Serum protein fractions albumin (Alb), alpha 1 (α1), alpha 2 (α2), beta 1 and beta 2 (β), and polyclonal gammaglobulins (γ) were quantified after pull down and graphed as percentage of control (lanes 2 and 6 in **f**). CR serum (top): AptC 'pulled out' CR M-Ig from serum but did not affect other CR serum protein fractions. Scrambled aptC (scr) had no effect. Control MM serum (bottom): AptC and scr control did not affect the concentrations of an unrelated M-Ig or serum protein fractions in control serum (Ctrl MM). All data represent means of at least $n = 3$ biologically independent samples ± SD, two-tailed Student t- test. **$p < 0.01$; ***$p < 0.001$; ****$p < 0.0001$.

complete remission before relapse illustrates the potential of using M-Ig specific aptamers for MRD monitoring. AptC detected residual disease in the first available sample obtained 6 months after ASCT. Concentrations gradually increased until MRD was detected by standard methods at 25 months (Fig. 4d). Given that all analyses were performed with neat serum, it is perceivable that sensitivities can be further increased by concentrating serum prior to analysis.

The overall process of aptamer isolation, assessment of binding affinity, and clinical validation can be standardized and be very cost-effective when compared to current methods. Low volumes of serum (less than 5 ml) are required to isolate sufficient M-Ig for SELEX and validation. Once isolated, aptamers can be reproduced without batch-to-batch variation and adapted to multiple platforms available in a clinical laboratory. Preliminary data (not shown) indicate that the same aptamers are functional in lateral-flow based point of care instruments. Historical samples can be added as a quality control and quantification standard, but there is no need to use historical samples to obtain a qualitative result.

Ultimate confirmation requires larger, prospective, and ideally multi-center validation studies incorporating multiple aptamers for monitoring to account for the possibility of clonal evolution[38]. The proof-of-principle study demonstrates the feasibility of isolating DNA aptamers that are specific to patient M-Ig, and their utility for serial M-Ig monitoring in serum at sensitivities required to detect MRD.

## Methods

**Commercial antibodies.** Monoclonal antibodies (mAb) daratumumab (anti-CD38 mAb) and rituximab (anti-CD20 mAb) were obtained from Johnson & Johnson and Genentech, respectively. Polyclonal human IgG (cat # 12511) and goat anti-human IgG-HRP (cat # AP112P) were purchased from Sigma-Aldrich.

**Patient samples.** De-identified serum samples from patients with MM were obtained under a Columbia University Irving Medical Center Institutional Review Board-approved protocol (IRB # AAAR7250). These samples included the following three patient groups: (1) Daratumumab-treated (D1–D3): Samples from

three patients with IgG Kappa MM before and after at least 6 months treatment with daratumumab ($n = 6$). (2) Stable disease (SD): Sequential samples from one patient with MM IgG Kappa SD ($n = 3$). (3) Complete response (CR): Sequential samples from one patient with MM IgG Kappa who achieved CR after autologous stem cell transplantation (ASCT), and relapsed 25 months after ASCT (total $n = 8$: before ASCT ($n = 1$); during CR when SPEP, IFE, Kappa, and Lambda free light chains (and their ratio), albumin, LDH, beta 2-micoglobulin and gammaglobulins were all within normal limits and MPFC was negative (Table 1; $n = 5$); following relapse ($n = 2$). In addition, we used two equal random mixes of sera as controls: (1) No-MM: 10 individuals without history of MM; and (2) MM: 10 individuals diagnosed with MM.

**Serum protein electrophoresis (SPEP) and immunofixation electrophoresis (IFE).** Sera were electrophoresed on agarose gels according to the manufacturer's instructions (Helena Laboratories, Beaumont, TX). Concentrations of serum protein fractions (albumin, α, β, γ globulins) and M-Ig were determined in reference to total serum protein (g dl$^{-1}$) using a HELENA QuickScan 2000 densitometer and software.

**Systematic evolution of ligands by exponential enrichment (SELEX).** The randomized oligonucleotide SELEX library (40 mer randomized region flanked by two primer regions (5′-GGAGGGCTCTCGGGACGAC-(N)$_{40}$-ATCGCTGCTGAC-CAGTTGAG-3′), PCR primers (fwd 5′GGAGGGCTCTCGGGACGAC3′, rev 5′CTCAACTGGTCAGCAGCGAT3′) and rev 5′ biotinylated aptamers were obtained from Integrated DNA Technologies (IDT). PCR mix (1 ml) consists of 10X Taq DNA polymerase buffer with 15 mM MgCl$_2$ (100 µl), Taq DNA polymerase 5 U/µL (Thermo Scientific cat # EP0402) (5 µl); 10 mM dNTPs mix (Promega cat # U1511) (20 µl); molecular biology grade water (Corning cat # 46-000-CM) (865 µl); primers (100 µM) (5 µl each). PCR protocol: 1 cycle at 95 °C, 2′; N cycles of [95 °C, 15″; 60 °C, 30″; 72 °C, 45″], 1 cycle of 72 °C, 2′; hold at 4 °C. N cycles refer to the number of PCR cycles that can vary from 7 to 21. SELEX Buffer (SB) is PBS, 2 mM MgCl$_2$. Elution buffer is PBS, 2 mM EDTA. Amplicons were electrophoresed on 3% agarose gels using 0.5x TBE buffer (Corning cat # 46-011-CM) and visualized with ethidium bromide. Positive PCR control is an aliquot of the original randomized ssDNA library. Negative control is 'no-DNA'.

The SELEX solid-phase target protocol was adapted from Yang[33]. Target M-Ig (3 nmol) was immobilized on protein G beads (Santa Cruz Biotechnology, cat # sc-2002) by incubating 200 µl of a 1:1 slurry equilibrated in SB for 30 min followed by four washes with SB (400 µl). In each SELEX round the immobilized target was incubated for 30 min with SELEX library (1 nmol) in round 1, or recovered ssDNA (80 pmol) in following rounds. Unbound oligonucleotides were removed by washing five times with SB (800 µl). Binding sequences were eluted by incubating three times for 10 min with EB (400 µl). Each elution was assessed by amplifying 2.5 µl in 10 µl of PCR mix and visualized on 3% agarose gels. Successful elutions

| Table 1 Clinical characteristics of patient CR. | | | | | | | | | | |
|---|---|---|---|---|---|---|---|---|---|---|
| Months | M-Ig (g dl⁻¹) | Free Kappa (0.57–2.63 mg dl⁻¹) | Free Lambda (0.57–263 mg dl⁻¹) | Kappa/Lambda (0.26–1.65) | Albumin (3.5–5.5 g dl⁻¹) | Beta 2-microglobulin (1.1–2.4 mg L⁻¹) | LDH (135–1600 U L⁻¹) | IgG (700–1600 mg dl⁻¹) | Gamma fraction SPEP ('polyclonal') (0.8–1.8 g dl⁻¹) | MPFC |
| 0 | 2.9 | 13.2 | 0.74 | 17.8 | 4.4 | 2.8 | 130 | 3272 | 3.2 | positive |
| 6 | ND | 2.61 | 1.72 | 1.5 | 5.1 | 2.4 | 173 | 1229 | 1.3 | negative |
| 12 | ND | 1.88 | 1.78 | 1.1 | 4.4 | 2.4 | 172 | 1064 | 1.1 | NP |
| 14 | ND | 1.69 | 1.64 | 1.0 | 4.5 | 2.4 | 162 | 1004 | 1.2 | NP |
| 17 | ND | 1.93 | 1.72 | 1.1 | 4.5 | 2.4 | NP | 1058 | 1.2 | NP |
| 22 | ND | 1.86 | 3.4 | 0.5 | 4.2 | 2.4 | 197 | 1217 | 1.4 | NP |
| 25 | 0.5 | 2.43 | 1.94 | 1.3 | 4.6 | 2.5 | NP | 1044 | 1.0 | positive |
| 28 | 0.6 | 4.71 | 1.52 | 3.1 | 3.9 | 2.3 | 175 | 1269 | 1.3 | NP |

Monoclonal immunoglobulin (M-Ig), free kappa light chains, free lambda light chains, kappa / lambda ratio, albumin, beta 2 microglobulin, lactate dehydrogenase (LDH), quantitative immunoglobulin G (IgG), and gamma fraction measured by serum protein electrophoresis (SPEP), show that all parameters were within normal range at 12–22 months. At 25 months, M-Ig was detected and quantified by SPEP. Free kappa light chains were increased and minimal residual disease (MRD) was detected by multiparameter flow cytometry (MPFC). Normal ranges are displayed. *ND* not detected, *NP* not performed.

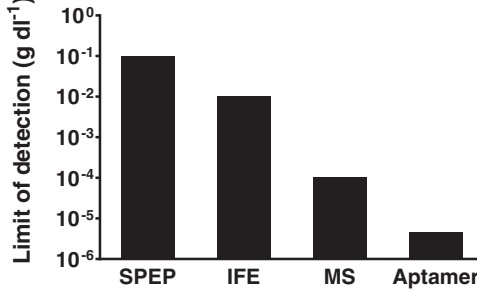

**Fig. 5 Comparison of sensitivity of serum-based approaches to measure M-Ig.** The LoD of M-Ig specific high-affinity aptamers are 2000-fold more sensitive than IFE.

were pooled and concentrated to 70 µl using a 10k MWCO centrifugal filter (Millipore Sigma cat # UFC501024), washed with water and added to a PCR mix containing 5′ biotinylated reverse primer (1 ml). An aliquot (100 µl) was used to test amplification at increasing PCR cycles (7–21 cycles). Optimal amplification was determined by amplicon migration on agarose gels and by the generation of sufficient DNA (80 pmol) for the consecutive SELEX rounds. The number of PCR cycles were recorded to monitor SELEX progress throughout the rounds. dsDNA strand separation was achieved by incubating the concentrated PCR products (10k MWCO centrifugal filter) for 10 min with streptavidin agarose resin (Thermo Scientific cat # 20353) (200 µl of 1:1 slurry washed and equilibrated in PBS), followed by three washes with PBS (800 µl) and by incubating with 250 µl of 0.2 N NaOH for 10 min. The flow-through containing ssDNA was neutralized with 0.2 N HCl, conditioned with 10 × PBS (Invitrogen cat # AM9624) to pH 7.4, then concentrated to 70 µl (10k MWCO centrifugal filter) and measured. Eighty pmol of ssDNA were used in the subsequent SELEX round. All procedures were carried out at room temperature.

Counter-selections (protein G immobilized polyclonal IgG, empty beads) were introduced by SELEX round 2. In these rounds, the flow-through was recovered, PCR amplified, strand-separated, and used for the next round. For more details, see Supplementary Fig. 1a. Finally, aptamers were cloned following the TOPO TA cloning kit protocol (Invitrogen, cat # K4500-01) and sequenced (genewiz.com). Sequences were analyzed for shared motifs with MultAlin (http://multalin.toulouse.inra.fr/multalin)[39] and for their secondary structures with Mfold (http://mfold.rna.albany.edu)[40].

Aptamers and scrambled controls are listed in Table 2. Primer sequences are underlined. Primer regions were not trimmed off as binding was found to decrease despite primers not being directly involved in it, but potentially providing additional structure for proper folding. Scrambled aptamers are aptamers in which the center between primer regions is randomized to control for non-specific binding by primer regions.

**Enzyme-linked oligonucleotide assays**. Two enzyme linked oligonucleotide assay (ELONA) formats were used, one where M-Igs were immobilized on protein G-coated plates and used to assess cross reactivity with other antibodies in buffer, and one where biotinylated target aptamers were immobilized on streptavidin-coated plates (pull-down format). This more stringent assay was used to demonstrate pull-down of the target M-Ig from serum[41].

Commercial antibodies or purified M-Igs (2.5 pmol, diluted in 100 µl of dilution buffer (DB; SuperBlock Fisher Scientific cat # 37535, 0.05% Tween 20, 2 mM MgCl₂)) were bound to protein G-coated plates (Thermo Scientific cat # 15156) for 30 min. The plates were washed eight times with wash buffer (WB; PBS 2 mM MgCl₂, 0.05% Tween 20) and then incubated for 45 min with 5′ biotinylated aptamer (25 pmol in 100 µl of DB). After washing ten times with WB, streptavidin-horseradish peroxidase (HRP) (Pierce high sensitivity streptavidin-HRP cat # 21230 (100 µl, diluted 1:30,000 with SB)) was added for 30 min, followed by washing 12 times with WB. TMB (3,3′,5,5′-tetramethylbenzidine) (Thermo Scientific cat # 34021) substrate was added and absorbance (370 nM) read 20 min later as well as in a time-course format every 5 min. Experiments were performed in triplicates for all data points and were expressed as means ± standard deviation.

5′ biotinylated aptamers (25 pmol in 100 µl of DB) were bound for 30 min to streptavidin-coated plates (Thermo Scientific, cat # 15124), washed eight times with WB and incubated for 1 h with 200 µl of blocking buffer (BB; ChonBlock Fisher Scientific cat # 50-152-6971). BB was discarded and the wells were dried. Control antibody (20 pmol), or serial serum dilutions in 100 µl antibody dilution buffer (AbDB; ChonBlock, 0.05% Tween 20, 2 mM MgCl2) were added to each well for 40 min, followed by washing ten times with WB. Goat anti-human IgG-HRP (Millipore, cat # AP112P) (100 µl, diluted 1:10,000 in AbDB) were added to each well for 30 min, then washed twelve times. TMB substrate was added and absorbance (370 nm) read 20 min later. Experiments were carried out at room temperature in triplicates for all data points and were expressed as means with standard deviations.

**Table 2 Sequences of aptamers and control aptamers.**

| | |
|---|---|
| AptD | 5′<u>GGAGGCTCTCGGGACGAC</u>GGCGCGGCGATTTGGGGTATGGGGAGGGGGTGGGTTGGGTCC<u>ATCGCTGCTGACCAGTTGAG</u> 3′ |
| Biotin-aptD | 5′BiosG/iSp18/<u>GGAGGCTCTCGGGACGAC</u>GGCGCGGCGATTTGGGGTATGGGGAGGGGGTGGGTTGGGTCC<u>ATCGCTGCTGACCAGTTGAG</u> 3′ |
| Biotin-apt D scrambled | 5′BiosG/iSp18/<u>GGAGGCTCTCGGGACGAC</u>GGTGCGTGCGGGGTCGGGTGGCGAGGTGGGGTAGTCAGTGGT<u>ATCGCTGCTGACCAGTTGAG</u> 3′ |
| AptS | 5′<u>GGAGGCTCTCGGGACGAC</u>CGTGGGCGGGGCGCAGAGGTAGGGTGTGGATGAGGGTGGGC<u>TATCGCTGCTGACCAGTTGAG</u> 3′ |
| Biotin-aptS | 5′BiosG/iSp18/<u>GGAGGCTCTCGGGACGAC</u>CGTGGGCGGGGCGCAGAGGTAGGGTGTGGATGAGGGTGGGC<u>TATCGCTGCTGACCAGTTGAG</u> 3′ |
| Biotin-aptS scrambled | 5′BiosG/iSp18/<u>GGAGGCTCTCGGGACGAC</u>GTGAGGCGGAGAGTGGCGCGTCGGAGGTGGACGGGTTGTGGG<u>ATCGCTGCTGACCAGTTGAG</u> 3′ |
| AptC | 5′<u>GGAGGCTCTCGGGACGAC</u>GAGTGTGTCAGGTGGATGGACGGTTGGGTTGGGTAGGCAAGG<u>ATCGCTGCTGACCAGTTGAG</u>3′ |
| Biotin-aptC | 5′BiosG/iSp18/<u>GGAGGCTCTCGGGACGAC</u>GAGTGTGTCAGGTGGATGGACGGTTGGGTTGGGTAGGCAAGG<u>ATCGCTGCTGACCAGTTGAG</u> 3′ |
| Biotin-aptC scrambled | 5′BiosG/iSp18/<u>GGAGGCTCTCGGGACGAC</u>TTAGGTTGAGAGGTGAGGCGTGGTGGATGGCGAGGCGTGGAT<u>ATCGCTGCTGACCAGTTGAG</u> 3′ |

**Surface plasmon resonance (SPR)**. Multi-parametric SPR (Bionavis MP-SPR Navi 210 A VASA) was used to assess the coefficient of dissociation ($K_D$). Biotinylated target and scrambled control aptamers were immobilized on regenerable SwitchAvidin sensors (BioNavis Regenerable avidin kit cat # SPR102-AVI) and assessed in two channels in parallel. SB was used as running buffer and injections were performed with either target (daratumumab or M-Ig) or control antibodies (human polyclonal IgG). All antibodies were diluted in SB. Samples were injected at increasing concentrations (6.25, 12.5, 25, 50, 100, and 200 nM) at flow rates of 20 µl/min. The duration of each injection and dissociation were 5 min, respectively. Results from the randomized sequence control channel were subtracted from the results of the aptamer interaction channel. Dissociation constants ($K_D$) were determined with the BIAevaluation software using a 1:1 interaction model.

**Homogeneous time resolved fluorescence (HTRF)**. This assay combines fluorescence resonance energy transfer (FRET) with time-resolved measurement (TR) to eliminate matrix inherent short-lived background to assess specificity and sensitivity of aptamers in serum. The donor fluorophore terbium (Tb) (ex. 340/em. 620 nm) (Cisbio cat # 610SATLF) was conjugated via streptavidin to biotin-labeled target aptamers. Allophycocyanin (d2) (ex. 620/em 665 nm) conjugated to goat polyclonal anti-human IgG-Fc (Cisbio cat # 61HFCDAF) was used as acceptor.

Patient or control sera (16 ul, diluted 1:10,000 with SB 0.05% Tween 20) were incubated overnight with terbium-labeled target aptamer or control (scrambled aptamer, no apt (=no DNA)) in a final volume of 20 µl (final concentrations 8 nM biotin-aptamer, 0.08 nM SA-Tb, and 6.7 nM IgG-d2) using 96-well low volume white plates (Cisbio cat # 66PL96100) that were covered with a polyester plate sealer (Corning cat # 4612). The HTRF ratio (665/620 nm) was measured on a Flexstation 3 (Molecular Devices) with a 50 µs integration delay and 300 µs integration time (ex 340/em. 620, em 665 nm). An increased HTRF ratio indicates that Tb-labeled aptamer is in proximity to d2-labeled anti-IgG Fc. All data points were carried out in triplicate at room temperature. Data shown are means with standard deviations.

**Pull-down assay**. Serum (50 µl) was incubated (1 h) with 2 nmol of biotin-aptamer, biotin-scrambled aptamer, or no aptamer control. Final volumes were adjusted to 57 µl with SB. Samples were added to columns packed with streptavidin-agarose beads and incubated for additional 45 min. Centrifugation for 2 min at 7000 g was done to collect the flow-through that was analyzed by SPEP and densitometry. The entire procedure was carried out at room temperature.

**Beads**. 800 µl of a 1:1 slurry of streptavidin agarose beads (Thermo Scientific cat # 20353) were washed 5 times with SB (1 ml). After the last wash, beads were suspended in SB (400 µl) of which 100 µl were dispensed into micro-spin columns (Thermo Scientific cat # 89879) that were centrifuged 2 min at 7000 g (Eppendorf 5415 D).

**Statistics and reproducibility**. Prism 8 software (GraphPad) and excel were used for statistical analyses. $P$ (two-sided) values are indicated in each figure. Sample sizes are defined in the legends and the text. Replicates are defined as independent samples. The LOD was determined using the standard deviation of the y-intercept and slope (LoD = 3.3 (standard deviation of the response/slope) of four independent experiments, each with a minimum of 6 concentration points performed in triplicate)[42].

**Reporting summary**. Further information on research design is available in the Nature Research Reporting Summary linked to this article.

## Data availability

The datasets generated during and/or analyzed during the current study are available from the corresponding author on reasonable request. Data underlying the graphs and charts presented in the main figures are available as Supplementary Data 1.

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

## Acknowledgements
We thank Milan N. Stojanovic, Dept. of Medicine and Dept. of Biomedical Engineering and Systems Biology, Columbia University Irving Medical Center for helpful discussions; the Columbia University Irving Institute for Clinical and Translational Research for a Pilot Award in Precision Medicine 'Personalized Aptamers for MRD in Multiple Myeloma (TSW). The Department of Pathology and Cell Biology for support (CTA, TSW). TRO received funding from the National Institutes of Health (UL1TR001873, TL1TR001875) and the Columbia Biomedical Engineering Technology Accelerator (BiomedX).

## Author contributions
C.T.A. and T.S.W. designed the studies. C.T.A. acquired and analyzed data. C.T.A. and T.S.W. wrote the manuscript; T.R.O. acquired and analyzed SPR data and revised the manuscript.

## Competing interests
The authors declare no competing interests.
