## [Peer Review File · Communications Biology]

Reviewers' comments:

Reviewer #1 (Remarks to the Author):

The author has done a cross-sectional study about detecting the minimal residual disease of multiple myeloma patients through M-Ig targeted aptamer. However, the quality of the manuscript is very poor. Thus, I recommend rejection to this manuscript

Comments to author:

1. Based on the current work, the author could not provide enough evidence to prove that selected aptamers indeed bind to targeted M-Ig protein in serum, only dependent on ELONA assay alone is not sufficient to support the author's conclusion, it should be validated by other approach. It is worth noting that once multiple myeloma patients received the treatment, many protein levels can be changed in serum, thereby the unspecific binding of aptamer to other proteins could influence the absorbance of aptamer bound M-Ig signaling.
2. More data should be provided to explain how to screen and characterize the selected aptamers.
3. Figure1-3 and supplemental figure 3 should be combined to one figure, while figure 4 and supplemental figure 5-6 should be combined to one figure. Concise typesetting could make the manuscript easier to understand.
4. In Fig.6A, the author showed that M-Ig were detected negative in the serum sample during 6-22 months, but detected positive at 25 months, however I could not find any significant difference in these two figures. Do I miss something?
5. In IFE data, full name of G,A,M,K, λ are needed in figure legends.

Reviewer #2 (Remarks to the Author):

The research is intended to develop a novel strategy for patient-specific multiple myeloma diagnostics platform using DNA aptamers. The results presented in the paper are original. Individual aptamer-based diagnostics platforms emerge more and more scientific interest and have a potential for wide practical implementation.

Aptamer selection process underlies this research. Moreover, authors intend to develop a universal aptamer selection protocol for M-Ig recovered from individual patient's blood. Respecting this, some questions can be addressed to the SELEX process and aptamer characterization:

- 1) More details on SELEX process are required. Is there any background for performing six selection rounds? Was it enough for efficient selection and why? Was the selection experiment assisted with progress evaluation?
- 2) More detailed data on sequencing and aptamer identification is required. Namely, the amount of sequences clones, frequency of identified sequences, reasons for selecting the aptamer sequence for further evaluation.
- 3) Aptamer-target binding characterization demands obligatory implementation of non-binding and no-DNA controls. In this paper, non-binding DNA control is mentioned for SPR experiment in Supplementary material. Please, specify the control sequence. No-DNA control experiment is also recommended, possibly using other methods besides SPR. Mention of control experiments should be added to main text due to its importance.

Selectivity and specificity tests as well as further diagnostics implementation of derived aptamers seem to be reliable. The overall result is promising for practical implementation.

Response to reviewers' comments on manuscript COMMSBIO-20-0908-A "Personalized immunoglobulin aptamers. A sensitive approach to measure multiple myeloma minimal residual disease in serum".

General response: We thank the reviewers for their thoughtful comments. We apologize for the lack of detailed experimental information and for not having shown or described results obtained with controls. We are addressing each concern and suggestion in detail below. The manuscript has been extensively revised to address and rectify all shortcomings.

Reviewer 1:

Comments to author:

1) Based on the current work, the author could not provide enough evidence to prove that selected aptamers indeed bind to targeted M-Ig protein in serum, only dependent on ELONA assay alone is not sufficient to support the author's conclusion, it should be validated by other approach. It is worth noting that once multiple myeloma patients received the treatment, many protein levels can be changed in serum, thereby the unspecific binding of aptamer to other proteins could influence the absorbance of aptamer bound M-Ig signaling.

Response: We agree with the reviewer. We addressed this central concern with two independent approaches. We developed a homogenous time-resolved fluorescence energy transfer (HTRF) assay to assess interaction of aptamers and targets in serum without the requirement for washing or immobilization, and carried out serum M-Ig pull-down experiments to demonstrate that aptamers bind to their target M-Ig in serum.

The homogenous time resolved fluorescence (HTRF) assay combines fluorescence energy transfer (FRET) between a donor fluorophore (here: terbium (340/620nm)) with an acceptor fluorophore (here: allophycocyanin (620/665nm)) with time-resolved measurements to eliminate short-lived background fluorescence. FRET is assessed, after excitation at 340 nm, by the HTRF ratio (665/620) and occurs when the fluorophores are in close proximity. In contrast to ELONA, the assay does not require washing and targets or aptamers are not immobilized. Therefore, all potential interferences, including other proteins generated or acquired during MM treatment that could have been removed during washing steps in ELONA, or affected by the ELONA format, are present at all times, for the entire duration of the assay. The HTRF format is detailed below.

Biotin-aptamers or control aptamers (biotin-aptamer scrambled) bound to terbium (streptavidin-terbium) were incubated in serum with acceptor-fluorophore-labeled goat anti-human IgG antibody. This antibody binds all IgG. FRET however only occurs when both molecules (terbium on aptamer and allophycocyanin on anti-IgG) are bound to the same M-Ig. FRET was only detected when target aptamers were incubated with target serum but not when incubated with control sera obtained from 10 patients with MM or 10 patients without MM (MM, no MM) (Figures 1F, 3C, 4E) or pre daratumumab treatment when testing aptD (Figure 1F).

The data support that the aptamers do not bind to another molecule in serum or a molecule that could have arisen during treatment.

To provide further evidence that the aptamers interact with their target M-Ig in serum, we performed pull down assays where biotin-aptamers were first incubated with serum, followed by

removal of the target M-Ig on a streptavidin column. The sera were then evaluated by SPEP. Results show that the aptamers decrease the target M-Ig without affecting other serum protein fractions, including the gamma fraction where polyclonal immunoglobulins migrate. Quantification by densitometry confirm the lack of cross-reaction with other serum proteins or cross-reaction with IgG M-Ig from a control patient. Randomized aptamers and no apt (=no DNA) conditions were included as controls (Figure 3D and 3E for the SD patient and Figure 4F and 4G for the CR patient).

2) More data should be provided to explain how to screen and characterize the selected aptamers.

Response: We agree with the reviewer and apologize for the negligence. We have now outlined in detail how aptamers were screened and selected and added these descriptions to the 'Materials and Methods' section and to the main text where a more detailed description was considered helpful. We added further details of the procedure per SELEX stage (Supplemental Figure 1A), provided examples for evaluation of SELEX progress (Supplemental Figure 1B and 1C) and included the frequency of the isolated sequences both in the text and in Supplemental figures (Supplemental Figures 1E, 3D and 3E).

3) Figure1-3 and supplemental figure 3 should be combined to one figure, while figure 4 and supplemental figure 5-6 should be combined to one figure. Concise typesetting could make the manuscript easier to understand.

Response: We agree with the reviewer and have made the suggested changes. The figures have been combined and merged into the new Figure 1 and new Figure 2, thereby increasing clarity significantly. We apologize for the typesetting. It has been changed to Arial.

4) In Fig. 6A, the author showed that M-Ig were detected negative in the serum sample during 6-22 months, but detected positive at 25 months, however I could not find any significant difference in these two figures. Do I miss something?

Response: The reviewer's point is valid and illustrates the limitations of SPEP in general to detect M-Ig at low concentrations. The particular SPEP gel (now Figure 4A) makes it difficult to distinguish differences by eye. Therefore, we included quantification obtained by densitometry and the immunofixation (IFE) gel (Figure 4A, bottom) that clearly shows the presence of the IgG Kappa at 25 months. The point has been included in the manuscript.

5) In IFE data, full name of G,A,M,K, λ are needed in figure legends.

Response: We apologize for the oversight. We have included the descriptions into the legends of Figures 3 and 4.

Reviewer 2.

The research is intended to develop a novel strategy for patient-specific multiple myeloma diagnostics platform using DNA aptamers. The results presented in the paper are original. Individual aptamer-based diagnostics platforms emerge more and more scientific interest and have a potential for wide practical implementation. Aptamer selection process underlies this research. Moreover, authors intend to develop a universal aptamer selection protocol for M-Ig

recovered from individual patient's blood. Respecting this, some questions can be addressed to the SELEX process and aptamer characterization.

General Response. We thank the reviewer for the positive assessment and for highlighting the potential of aptamers as diagnostics. We are addressing each concern and suggestion in detail below.

Reviewer 2 # 1. More details on SELEX process are required. Is there any background for performing six selection rounds? Was it enough for efficient selection and why? Was the selection experiment assisted with progress evaluation?

1a) Is there any background for performing six selection rounds?

Response: Yes, selection of high affinity aptamers after seven cycles¹⁻³ and even one cycle⁴⁻⁶ has been reported. Of note should be that the methods used by these groups are not exactly the same as the one we use, but rather a proof-of-concept that high affinity aptamers can be isolated at earlier SELEX rounds.

1b) Was it enough for efficient selection and why?

Response: We based our decisions on the data previously reported by other laboratories (please see above) and on our SELEX progress evaluation (by monitoring PCR cycles required to amplify the eluted binding sequences, Supplemental Figure 1D; 3B and 3C, please see answer to next question). An advantage of testing earlier rounds is that in the case of not finding high performing aptamers one can continue with the last pool of isolated sequences and further increase the rounds of SELEX. Alternatively, one can change the randomized oligonucleotide library or stringency.

1c) Was the selection experiment assisted with progress evaluation?

Response: Yes, and the criteria were included in Materials and Methods, the main text and Supplemental Figures 1D, 3B and 3C. For SELEX progress evaluation we used the number of PCR cycles required to amplify enough material to advance to the next round of SELEX (denoted by a visible band on a 3% agarose gel stained with ethidium bromide where the limit of detection is ≈ 5 ng). A generality is that the number of PCR cycles required to amplify the recovered binding sequences decreases as the number of SELEX rounds increases due to enrichment of binding sequences by PCR (Supplemental Figures 1C, 3B and 3C). The number of PCR cycles required for amplification per SELEX round are recorded and used to compare progress.

2) More detailed data on sequencing and aptamer identification is required. Namely, the amount of sequences clones, frequency of identified sequences, reasons for selecting the aptamer sequence for further evaluation.

Response: We added this information both in the main text and in Supplemental Figures 1E, 3B and 3E. In summary, we sequenced around 20 sequences per target. Criteria for selection were the binding to target in serum. The frequencies of selected aptamers for these criteria were as follows. For the daratumumab and patient SD SELEX (Supplemental Figure 1E and 3D) about 37% of the total isolated sequences were the same aptamer that is described in the

manuscript. These were selected for further studies based on their ability to bind the target in serum. From the daratumumab SELEX we isolated only one more sequence that we are currently using for a different clinical application and therefore not describing here. Of the other sequences isolated during the patient SD SELEX, 21% bound to the target M-Ig only in buffer and 42% did not bind in any condition. For patient CR, as described in the main text and Supplemental Figure 3E, 10% of the sequences were the aptamer described in this paper, 50% was one sequence that bound to the target only in buffer and 40% did not bind to the target in any condition. This information has been included in Supplemental data.

3) Aptamer-target binding characterization demands obligatory implementation of non-binding and no-DNA controls. In this paper, non-binding DNA control is mentioned for SPR experiment in Supplementary material. Please, specify the control sequence. No-DNA control experiment is also recommended, possibly using other methods besides SPR. Mention of control experiments should be added to main text due to its importance.

Response: We agree with the reviewer. This information, including that of control sequences that consist of scrambled non-primer regions, were added to Materials and Methods, the main text and figure legends. We show the no apt (=no DNA) control and scrambled aptamer (aptamers in which the sequences between primer regions are randomized (Table 1)) in all assays. Primers were not trimmed off as binding was decreased despite primers not being directly involved in the binding, potentially providing additional structure for proper folding.

- 1 Mencin, N. et al. Optimization of SELEX: comparison of different methods for monitoring the progress of in vitro selection of aptamers. *J Pharm Biomed Anal* 91, 151-159, doi:10.1016/j.jpba.2013.12.031 (2014).
- 2 Levine, H. A. & Nilsen-Hamilton, M. A mathematical analysis of SELEX. *Comput Biol Chem* 31, 11-35, doi:10.1016/j.compbiolchem.2006.10.002 (2007).
- 3 Zamay, G. S. et al. Development of DNA Aptamers to Native EpCAM for Isolation of Lung Circulating Tumor Cells from Human Blood. *Cancers (Basel)* 11, doi:10.3390/cancers11030351 (2019).
- 4 Berezovski, M. V., Musheev, M. U., Drabovich, A. P., Jitkova, J. V. & Krylov, S. N. Non-SELEX: selection of aptamers without intermediate amplification of candidate oligonucleotides. *Nat Protoc* 1, 1359-1369, doi:10.1038/nprot.2006.200 (2006).
- 5 Lou, X. et al. Micromagnetic selection of aptamers in microfluidic channels. *Proc Natl Acad Sci U S A* 106, 2989-2994, doi:10.1073/pnas.0813135106 (2009).
- 6 Liu, Y. et al. DNase-mediated single-cycle selection of aptamers for proteins blotted on a membrane. *Anal Chem* 84, 7603-7606, doi:10.1021/ac302047e (2012).

REVIEWERS' COMMENTS:

Reviewer #1 (Remarks to the Author):

The author in the manuscript have done great work about constructing a new method for detecting the MRD of multiple myeloma patients through personalized M-Ig targeted aptamer. Based on the previous comments, the author carefully modified the manuscript and reorganize the figures. Thus, I recommend the acceptance of this manuscript.

Reviewer #2 (Remarks to the Author):

The research is intended to develop a novel strategy for patient-specific multiple myeloma diagnostics platform using DNA aptamers. Individual aptamer-based diagnostics platforms emerge more and more scientific interest and have a potential for wide practical implementation. The results presented in the paper are original. The methods are described well to be easily reproduced. The results are well validated. Taken together, the originality, novelty and scientific quality of this research, as well as the quality of paper preparation, make it possible to recommend the manuscript for the publication in the present form.